# Method Development for Detecting Low Level Volatile Organic Compounds (VOCs) among Workers and Residents from a Carpentry Work Shop in a Palestinian Village

**DOI:** 10.3390/ijerph20095613

**Published:** 2023-04-23

**Authors:** Shehdeh Jodeh, Abdelkhaleq Chakir, Ghadir Hanbali, Estelle Roth, Abdelrahman Eid

**Affiliations:** 1Department of Chemistry, An-Najah National University, Nablus P.O. Box 7, Palestine; 2Groupe de Spectrométrie Moléculaire et Atmosphérique GSMA, UMR CNRS 7331, Université de Reims, Moulin de la Housse B.P. 1039, CEDEX 02, 51687 Reims, France; 3Department of Mathematics, An-Najah National University, Nablus P.O. Box 7, Palestine

**Keywords:** volatile organic compounds, solid phase micro extraction, paint, analysis, blood

## Abstract

Volatile organic compounds (VOCs) are considered a major public health concern in industrial location areas. The presence of exposure to (VOCs) has raised concern regarding the health effects caused by chronic human exposure as this will increase cancer diseases in the village. An analytical method has been developed and modified to help us detect 38 VOCs in the blood of 38 volunteers who are related to a carpentry shop at the parts-per-trillion level. To measure and evaluate the potential risk, several devices, such as portable passive monitors and air-collected samples, in addition to blood concentration, were used to study three different occupational groups. Ten of the volunteers are employees at the shop, 10 volunteers live very close to the shop, and 10 of them are students in an elementary school very close to the shop. In this study, we developed an automated analytical method using headspace (HS) together with solid-phase microextraction (SPME) connected to capillary gas chromatography (GC) equipped with quadrupole mass spectrometry (MS). The detection limits for the method used were measured in the range from 0.001 to 0.15 ng/L, using linear calibration curves that have three orders of magnitude. The detected concentrations ranged from 3 ng L^−1^ for trichloroethene to 91 ng L^−1^ for toluene and 270 ng L^−1^ for 2,4-diisocyanate, which was derived from the paint solvents used for the wood in the carpentry shop and the paints on the walls. More than half of all assessed species (80%) had mean concentration values less than 50 ng L^−1^, which is the maximum allowed for most VOCs. The major chemical types among the compounds quantified will be those we found in our previous study in the surrounding air of a carpentry workshop in Deir Ballout in Palestine, which were toluene diisocyanate and butyl cyanate. Some were found to be highly present air. Most of the measurements were below the guidelines of the World Health Organization (WHO). Despite the fact that this study only involved a small number of smokers, smoking was found to be connected with several blood and breath components. This group includes unsaturated hydrocarbons (1,3-butadiene, 1,3-pentadiene, 2-butene), furans (2,5-dimethylfuran), and acetonitrile. The proposed classification of measured species into systemic (blood-borne) and exogenous volatiles is strictly hypothetical, as some species may have several origins.

## 1. Introduction

Volatile organic compounds (VOCs) can be found in almost any house or workplace, and they come from both natural and man-made sources. Health issues were also raised after utilizing plywood, particle board, and fiberboard in indoor situations that had adhesives containing formaldehyde in their compositions. Formaldehyde is generated by wood products collected in restricted spaces at amounts greater than those found in ambient air. The IARC classifies formaldehyde as a human carcinogen (Group 1), and it is a common contaminant in non-industrial indoor environments [1]. Building ventilation and maintenance have been and continue to be the focus of study in terms of indoor air quality, as has the topic of how much the outer environment affects or modifies the air quality inside enclosed spaces through the penetration of contaminants from outside [2].

Hazardous volatile organic compounds (VOCs) such as benzene, toluene, ethylbenzene, and xylenes (BTEX) are air pollutants that endanger human health [3]. VOC ingestion, inhalation, and skin exposure can all result in a variety of negative health impacts. Because of the volatile nature of VOCs, inhalation is the most common route of exposure. Previous research has shown that inhaling VOCs increases the chance of developing certain diseases, including asthma [2,3], cardiovascular disease [3,4], and chronic obstructive pulmonary disease (COPD). However, not all VOCs breathed enter the respiratory system. The human body can pollute indoor air by generating volatile organic compounds (VOCs) through breath and skin [4]. However, studies on the quantitative characteristics of skin VOC emissions are sparse when compared to those on the quantitative characteristics of breath emissions. Furthermore, the species and rates of VOC emission from a single person’s breath and whole-body skin have rarely been documented. Prior research on the emission characteristics of VOCs from industrial sources in the United States and elsewhere has primarily concentrated on industries such as the chemical industry, oil refineries, and solvent use [5].

As a result, the percentage of harmful VOCs breathed that are deposited and absorbed inside the body is essential in terms of health concerns [4,5,6]. Unfortunately, alterations in VOC deposition rates dictated by subjects have received minimal attention. So far, Huang et al. have discovered that physiological factors such as gender, BMI, and body fat ratio are not the most relevant determinants determining Ash deposition rates [7,8,9,10,11]. Much research has been carried out to highlight the influence of subjects on VOC deposition rates. For example, changes in breathing models have an instantaneous effect on exhaled VOC concentrations [7,8,9,10,11]. While the quantitative study of breath components has garnered extensive interest, few studies have been conducted to investigate the levels of these volatiles in human blood [12,13].

Volatile organic compounds (VOCs) are a major public health concern in the developed world.

Many critical aspects remain unanswered in determining exposure to these chemicals.

Because VOCs are ubiquitous and very volatile, specialized procedures must be used in their analysis.

The analytical methodology used to evaluate toxicants in biological materials must be adequately verified and meticulously carried out; inadequate quality assurance can result in inaccurate results that have a direct impact on treating exposed individuals.

The pharmacokinetics of VOCs demonstrate that the majority of the internal dose of these compounds is rapidly cleared, while a portion is only slowly eliminated, and these chemicals may bioaccumulate. VOC levels in the general population are in the high parts per trillion range, but some people with significantly higher levels appear to have been exposed to VOC sources outside of the workplace.

When analyzing suspected cases of VOC exposure, smoking is the most significant confounder of internal dosage levels.

These approaches must be used with caution because contamination and analyte loss are still major concerns, but they are practical and have been used by numerous research groups. VOC levels in the body alter fast during exposure and after cessation of exposure. Most VOC internal dose levels decline rapidly after exposure ends, with most having a half-life of a few hours; however, the actual decrease depends on the VOC.

Furthermore, most blood VOC research has focused on specific species (for example, harmful or carcinogenic chemicals) or groups of people (e.g., mechanically ventilated patients, smokers, or volunteers exposed to specified levels of pollutants).

Several investigations [14,15,16] on the blood levels of halogenated hydrocarbons and aromatics (BTEXS) as indicators of environmental exposure have been carried out.

In non-workplace exposure conditions [17]. Perbellini et al. (measured 1,3-butadiene, benzene, and 2,5-dimethyl furan levels in blood and breath. There are several sources of volatile organic compound (VOC) emissions both indoors and outdoors. However, the most common source of VOC exposure remains indoor air. The general populace is constantly exposed to harmful compounds because the bulk of people spend 90% of their time indoors (at home, school, or office).

Organic compounds are commonly used in household products as ingredients. Organic solvents are found in paints, varnishes, and wax, as well as numerous cleaning, disinfecting, cosmetic, degreasing, and hobby items. Organic compounds are used to make fuel. Any of these products can emit organic molecules when in use and, to a lesser extent, while being stored.

Chloroform can be poisonous to the liver for a long time.

Benzene, a common petrochemical and paint solvent, can reduce the number of red blood cells in the body, resulting in anemia and genetic damage [18]. Toluene can be broken down into hazardous chemicals; 5% of its metabolites are benzaldehyde and cresols Y. The majority of reactive compounds are detoxified by conjugation to glutathione, but the remainder may cause severe cell damage [19]. Important petrochemicals, ethylbenzene and xylenes, can irritate the nose and throat when inhaled, causing symptoms of central nervous system depression and harming the liver and kidneys [20]. Construction materials (for example, coatings, adhesives, paints, and sealants), furniture (for example, stains, varnishes, and fabric materials), and cleaning chemicals are all significant sources of indoor air pollution [21]. Indoor VOC exposure has been related to a number of acute and chronic health effects, including eye, nose, and throat irritation, asthma exacerbation (particularly in young children), allergic reactions, respiratory disorders, liver and kidney dysfunction, and cancer [19,20,21,22]. VOC levels in the indoor air of residences [23], schools, businesses, and commuting vehicles [24] are monitored to assess human exposure to VOCs. Traditional oil-based stains with higher VOC levels are also harmful to the skin and must be removed from brushes using paint remover or other chemical agents. You have probably heard that applying wood stains in well-ventilated areas is crucial [25]. The primary goal of this study was to characterize the broadest possible range of volatile organic compounds produced by solvents in paint that carpenters are regularly exposed to.

In this study, a method for evaluating 38 VOCs in blood samples at parts-per-trillion levels (ng/L) using SPME-GC-MS with single-reaction monitoring was developed (SRM). Regarding this method’s uniqueness, previous approaches for measuring VOCs in blood have lacked sensitivity, ruggedness, productivity, or broad application [26,27,28,29]. As a result, we devised an improved approach for quantifying 38 target VOCs in 3 mL of whole blood by combining headspace solid-phase microextraction (SPME) with capillary gas chromatography (GC) and quadrupole mass spectrometry (MS).

The novelty of this study is to obtain data from different age groups who are exposed due to the gases generated from the solvents of the paints; furthermore, it is the first study of this kind to be carried out in Palestine.

## 2. Materials and Experimental

### 2.1. Study Area

For the first time in Palestine, this study explores the concentration of VOCs in the blood of employees, students, and residents as a source of pollution concentrations indoors in a carpenter’s workshop in a community situated between residential buildings and a school building (Deir Ballout, Palestine).

Deir Ballout is a Palestinian town in the northern West Bank’s Salfit Governorate, 41 km south of Nablus. It has a population of 4650. The village is located 236 m above sea level and is surrounded by various Israeli colonies and an industrial zone, as shown in Figure 1.

Outside of the village, people are working on structures and buildings.

Twenty percent of the villagers are over the age of 70, and some of them work in agriculture. In total, 40% are involved in education and health, as well as military service in Palestine. The rest are schoolchildren.

The majority of residents in the hamlet possess cars, which may contribute to pollution in the village’s air. The percentage of divorced women is less than 0.1%, and the entire town is Muslim.

### 2.2. Materials

Several chemicals and solvents were acquired from Sigma-Aldrich (St. Louis, MO, USA) and utilized to manufacture all of the standards and rinse the glassware. J.T. Baker supplied HPLC-grade water (Phillipsburg, NJ, USA). Because water is frequently contaminated with varying trace quantities of VOCs, particularly chloroform, the source water was further cleansed using helium purging and distillation [30]. Because water purity varies greatly between production lots, enough water was distilled to ensure that the same lot was utilized to generate all solutions, blanks, standards, and quality control (QC) material for these investigations. Becton Dickinson supplied stainless steel needles (18-gauge, Luer-Lok) (Franklin Lakes, NJ, USA). Microliter Inc. provided reaction vials (10 mL, serum type) (Suwanee, GA, USA). Supelco supplied septa (20 mm, Teflonfaced/silicone), seals (aluminum, open center), SPME fibers (75 mCarboxen/PDMS), and SPME-GC inlet liners (0.75 mm I.D.) (Bellefonte, PA, USA). A GC Capillary Column Stabilwax 60 m, 0.32 mm ID, 0.25 m) was used for chromatographic separation (Restek, PA, USA).

### 2.3. Standards Calibration

Table 2 shows the standard sets that were purchased from Sigma Aldrich USA. Each calibration set had seven concentration levels: calibration solutions (0.010, 0.030, 0.100, 0.300, 1.00, 3.00, and 10.0 g/L) were prepared on the day of analysis by diluting each calibration standard in VOC-free water. Methanol was purged and trapped to make parent, intermediate, and working solutions, which were then transferred to flame-sealable glass ampoules, sealed, and kept at −70 °C until use. Aqueous solutions were made in VOC-free water on the day of analysis. Individual parent stock solutions were prepared in the same solvent as the internal standards. Intermediate and working stock solutions, including internal standards, were prepared using the same solvent.

### 2.4. Instruments and Apparatus

An autosampler (Combi-PAL, CTC Analytics AG, Zwingen, Switzerland) and a gas chromatography (HP 6890, Agilent Technologies, Palo Alto, CA, USA) equipped with a mass-selective detector were used for HS (HP 5973N, Agilent Technologies, Palo Alto, CA, USA). The injection was performed in split mode (split ratio 100:1) into fused silica (GC Capillary Column Stabilwax 60 m, 0.32 mm ID, 0.25 m) (Restek, PA, USA). Mass spectra were obtained at 70 eV in the electron ionization mode. The spectrometer was operated in the selected ion monitoring (SIM) mode.

Only full-scan mass spectra (m/z 35-450) were obtained to identify the analytes.

The temperatures of the source and analyzer were 230 and 150 °C, respectively. Total ion chromatograms were obtained and processed using Agilent Technologies G1701DA D.01.02 standalone data analysis software, which was also utilized to manage the entire system.

Supelco supplied the SPME gadget and fiber (Bellefonte, PA, USA).

The 10 mm long microextraction fiber was coated with a CAR/PDMS (polydimethylsiloxane) layer that was 85 m thick.

Before being used in the analysis, the new fiber was conditioned for 1–2 h in a GC injector port at 300 °C. The conditioned fiber was utilized right away or was kept clean by putting the SPME syringe needle into a GC septum injection port before use.

Sampling was optimized by sampling for 2 min, stirring at 500 rpm, and desorption at 250 °C for 0.5 min in 11 mL vials with 2 mL sample volumes (1 mL whole blood and 1 mL water).

Then, the 2 mL samples were inserted in 4 mL glass vials with silicon septum closures, spiked with VOCs, and adjusted to pH 11 using 0.4 g sodium chloride and 0.04 g potassium carbonate anhydride.

Whole-blood samples were spiked with VOCs and combined with sodium chloride and HCl, citric acid, K_2_CO_3_ or KOH for pH control in 10 mL glass vials with silicon septum closures.

Capped vials were inserted in the autosampler. Headspace was performed after 30 min of vial equilibration at 60 °C and 2 s stroke times for injection volume at 90 °C. To determine the best conditions, the samples were agitated in sealed vials for 20, 30, 40, 50, and 60 min at 30–70 °C. When the VOCs had reached equilibrium, the needle was inserted into the sealed sample vial, and the heated gas-tight syringe was used to pump the headspace twice. The syringe was immediately withdrawn from the vial and put into the GC injection port at 200 °C. After that, it was returned to its original spot and cleaned with flush gas [31].

### 2.5. Sampling Sites

Indoor air samples were collected from 4 locations: the first one is the carpentry shop, the second one is a residential house located west of the shop within 20 m, the third location is another resident house north of the shop within 40 m, and the last location is about 200 m north of the shop, and it is an elementary school and all of them are located in the village of Deir Ballout. During the initial visit, we provided questionnaires to the residents in order to collect information about the built environment, such as building attributes, interior decorating, time–activity patterns, and household lifestyles.

In this investigation, collecting from outdoor samples was not practicable due to a lack of power supply and secure setup locations for sampling devices in the household’s outdoor areas.

### 2.6. Field Sampling and Chemical Analysis

EPA TO-17 was utilized in our investigation to collect air samples and evaluate the examined VOC components in indoor air.

Passive VOC samples were collected in thermal desorption adsorbent tubes directly (Sigma Aldrich, St. Louis, MO, USA). Jia et al., 2008 outline the approach and its performance in greater detail [10]. The indoor air sample, which lasted 24 h, was undertaken in each residence’s living room. The sampling apparatus was set as centrally as practicable, with the inlet 1.0–1.5 m above the ground (breathing zone). The sample rate was set to ten milliliters per minute. As a consequence, the total volume of the sample was 14.4 L. Meanwhile, we assessed the temperature, relative humidity (RH), and air exchange rate (AER) in the indoor environments during sample intervals. The 24 h average internal temperature and RH were 23.3 ± 0.4 °C and 51.6 ± 14.5%, respectively. In these houses, the average AERs were 0.68 ± 0.50 h^−1^.

The tubes were delivered to the laboratory after sampling and evaluated within 5 days using an automated thermal desorption system (Scientific Instrument Services, Inc., Ringoes, NJ, USA), followed by the gas chromatography/mass spectrometry described in Section 2.4. For duplicate samples, a novel analytical method was used: one tube was analyzed in the MS scan mode and the other in the selected ion monitoring (SIM) mode to take advantage of the SIM mode’s high sensitivity. These approaches produced comparable results, while the SIM mode produced lower method detection limits (MDLs) of 0.002–0.27 µg m^3^, depending on the substance, compared to the scan mode’s MDLs of 0.015–0.51 µg m^3^. The quality assurance/quality control (QA/QC) procedures for sample collection, transportation, storage, handling, and analysis are detailed in detail elsewhere.

After being stored in a refrigerator at 3 °C for one week, all of the samples were evaluated within one week. For both carbonyl and non-carbonyl VOCs, two standards were created. The VOCs standard solution (1000 g/mL in methanol: water 97:3, Supelco Inc., Bellefonte, PA, USA) was diluted with methanol (HPLC grade, Sigma Aldrich, St. Louis, MO, USA) to generate standard solutions of varied concentration grades for non-carbonyl species.

The working standards were then created by inserting a 1 L aliquot of each grade of the liquid standard into the adsorption bed of the pre-conditioned tubes using a 10 L GC syringe. As a result, tubes with mixed VOC standards of 1, 2, 5, 10, 20, 50, and 100 ng were created. Thermal desorption (Markes International LTD, Llantrisant, UK) coupled with GC–MS in a single split and selective ion mode was used to analyze all of the samples and working standards.

### 2.7. Human Subjects and Sampling

This study is based on the approval of the families of the students in the elementary school. Additionally, a special form was filled out and approved by the Institutional medical board of the university.

A group of 30 individual volunteers was chosen to represent our study with various ages and genders (Table 1). Ten males work daily in the carpentry shop. Five of them are heavy smokers. Ten people from the residents are not more than 30 m away from the carpentry shop. Two of them are smokers, and 5 of them are females. Ten students study in the elementary school, which is located 200 m away from the carpentry shop. The average age is between 12 and 16 years old, while the average of employees is 27 years. We labeled their names as follows: The first 3 letters designated for first, median, and last name. M or F designated gender. S and NS indicate smokers and nonsmokers, and finally, the number is the age of the volunteer (Table 1). All of the subjects will provide written informed consent to participate. There will be no explicit meal limitations; however, volunteers will be asked to rest for at least 10 min before sampling in order to avoid fluctuations in temporal breath VOC concentrations caused by activities [32,33]. Furthermore, prior to the sample stage, all participants will remain in the room for at least one hour.

During this period, participants will fill out a questionnaire concerning their health, smoking habits, and recent food intake. Each participant will have their venous blood sampled twice. This will be undertaken to identify potential pollutants that are not derived from the blood. The concentration levels obtained from the blank samples will be deducted from the blood sample readings.

### 2.8. Blood Collection Vial Preparation

Table 1 summarizes the participants in the study. The participants’ blood samples were taken for VOC analysis in specially prepared blood collection vials (Vacutainers^®^, Becton-Dickinson, Franklin Lakes, NJ, USA) with an 8 mL draw (13 mm 100 mm). Depending on the needs of the investigation, vacutainers were utilized. These gray-top vacutainers contain potassium oxalate and sodium fluoride to prevent clotting and slow cellular metabolism. The vacutainers were disassembled, and the butyl rubber stoppers were roasted in a vacuum oven (100 kPa) for 17 days at 80 °C to eliminate VOC residue, as reported by Cardinali et al. [33,34]. The vacutainer vials were also roasted in a vacuum oven at 80 °C one day before reassembly (100 kPa). The ovens were equilibrated to atmospheric pressure with ultrahigh-purity nitrogen after cooling to ambient temperature. After that, the vacuum was redrawn with a small-gauge needle (1/2-in. aluminum Luer lock hub, 27-gauge, Sherwood Medical, St. Louis, MO, USA).

Following the restoration of suction, the vacutainers were disinfected using an AECL Gamma Cell 220 Irradiator with a Cobalt-60 source. To inactivate any germs that may have been introduced into the vacutainer during processing, it was dosed with roughly 250,000 rads. The shelf life of vacutainers prepared using this technology is one year.

### 2.9. Method Validation

VOC levels were evaluated in unknown samples using calibrators ranging from 0.01 to 10 g/L (Table 2).

A large linear range was required due to the substantially changing VOC levels in the human blood samples; prior research has shown that VOC levels in human blood can vary by more than two to three orders of magnitude between different people depending on their health [35,36]. For each VOC, each sample group was tested using at least seven calibrators, and the relative response of those calibrators was utilized to generate the calibration curve for that time of concentration measurements.

Calibration curves are used in conjunction with the concentration inverse and frequently have correlation values greater than 0.99, with strong linearity across the calibration curve at low concentrations. LODs were computed using extracted ion chromatograms and the standard deviation of 10 consecutive blank signals, as previously undertaken in other studies [37,38].

As blanks in the case of blood species, conditioned human plasma samples were used. Blood LOD values ranged from 0.01 to 280 nmol L^−1^. Figure 2 shows a chromatogram from a blood HS-SPME-GCMS analysis.

### 2.10. Quality Assurance

The data were subjected to quality control methods utilizing a proprietary laboratory information management system designed in Microsoft Access.

Contamination was evaluated both subjectively and objectively. SPME was used to collect laboratory air for 6 min before analyzing it with SPME-GC-MS, as previously mentioned. The resulting chromatograms were qualitatively examined for VOC contamination.

A VOC-free water blank was also evaluated for analyte contamination that could affect the results. We looked at additional QC criteria after examining the samples and visually inspecting each integrated peak.

The absolute peak area signal and signal-to-noise ratio were used to assess acceptable labeled analog response.

We further confirmed the analyte ion identification by comparing the confirmation ion ratio in unknown samples to that of reference standards. Each batch of data was compared to unblinded QC samples.

### 2.11. Quality Control Samples

For each batch of samples, four QC samples were processed and evaluated.

The VOCs were equilibrated in a mixture of concentrated standards and bovine serum to create these samples. Aliquots were then stored in flame-sealed glass ampoules at 70 °C.

An aliquot of QC serum was taken on the day of use as though it were an unknown. An unbiased QC officer evaluated blind QC samples using modified Westgard QC standards [35,36]. At least 15 independent measurements were used to determine the assay accuracy of each analyte.

If the results differed from specified means by three standard deviations, two consecutive measures exceeded two standard deviations, or ten consecutive readings fell on the same side of the mean, QC failed for that analyte.

If an analyte’s QC sample results failed, all findings for that analyte on that day/batch were deleted.

### 2.12. Blank Analysis

In a typical laboratory, trace levels of VOCs such as methylene chloride, chloroform, trimethylsilanol, benzene, toluene, xylenes, and methyltert-butyl ether (MTBE) are frequent; rigorous protocols are required to reduce sample contamination from laboratory air.

Laboratory air supplies have been proven to be polluted by chlorinated water, common domestic cleaning chemicals, laboratory solvent use, and exhaust from oxygenated fuel consumption.

These and other volatile pollutants can easily cross-contaminate samples during preparation (sample handling) or analysis via laboratory air (SPME fiber). Contamination was reduced by removing VOC sources from the laboratory (where possible). A blank water sample was used to test for contamination. Helium sparging, distillation, and flame sealing were used to make blank water in glass ampoules. On the day of use, a water blank was spiked with specified internal standards and run with each batch of unknowns. The run was marked as contaminated for that analyte if the analyte levels in the blank exceeded the LOD. An SPME fiber sample of laboratory air was also taken to qualitatively assess airborne contaminants.

### 2.13. Proficiency Testing

A commercially available VOC mixture was utilized to create competence test items (Environmental Protection Agency mix 524 rev A, Supelco).

Additional analytes not identified in this VOC combination, such as 2,5-dimethylfuran, were fortified with gravimetrically validated amounts of pure material. Through successive dilution into methanol, individual proficiency testing pools were generated.

Each proficiency testing pool was aliquoted and flame sealed into glass ampoules.

An independent QC officer blind-coded these ampoules, and five ampoules were evaluated and blinded every 6 months or after severe instrument repair.

The proficiency testing components were diluted in distilled water and analyzed on the day of analysis. If the blind evaluated levels were within 25% of the true values, the assay passed proficiency testing.

The most difficult aspect of assessing VOCs is collecting samples and preparing samples, particularly VOC in blood. Gas chromatography Mass spec is being used for analysis. However, GC-MS has two fundamental limitations: first, only a limited range of volatile, thermally stable chemicals may be analyzed, and second, mass spectra frequently lack or have weak molecular ions.

A photoionization detector, or PID, is the most frequent equipment used by professionals to measure VOCs in a property. These instruments are typically portable and estimate the overall number of VOCs in the air. Although laboratory analysis is the most convincing method of measuring VOCs, it is not always the best option due to the cost and turnaround time. Several labs can assess a sample for a panel of hundreds of VOCs, which can be quite useful when the irritant is unknown, or there are numerous target chemicals. Certain VOCs, such as formaldehyde, require specialized laboratory testing and are not detectable by panel analyses. Typically, samples are collected on sorbent tubes, passive badges, or evacuated canisters. The air quality professional will determine the optimal sampling method and laboratory analysis based on the specific situation. GC-IMS detected a wide range of volatile chemicals, primarily oxygen-containing VOCs such as alcohols, fatty acids, aldehydes, and ketones, with low odor threshold values. The fingerprint plot of IMS spectra revealed variations in the composition of VOCs, indicating changes in wastewater quality during the treatment process. The GC-IMS approach may provide an efficient profiling method for changes in inlet water and treatment process performance at WWTPs.

## 3. Results and Discussion

### 3.1. Summary of Data

Table 2 summarizes the results of the samples collected from the study area, which included a school, residential dwellings, and a carpenter shop.

A Total Ion Chromatogram (TIC) is a graphical representation of the total ion intensity detected at each point in time during a GC-MS examination. A TIC can be used to display the total amount of VOCs discovered in a sample over time in the context of VOCs (volatile organic compounds).

When GC-MS analysis of a VOC-containing sample is performed, the sample is first injected into the GC column and separated into its various components as it moves through the column. Once each component departs from the column, the mass spectrometer ionizes it and detects it as a distinct mass-to-charge (m/z) ratio. Based on its chemical characteristics and interactions with the stationary phase in the column, each VOC in the sample would elute from the GC column at a different retention time.

The peak height and width of the VOC-corresponding TIC peak can provide information about the VOC’s concentration and retention duration in the sample. The TIC can also be used to compare the overall number of VOCs discovered in various samples, as well as to identify and quantify several VOCs present in a single sample. Figure 2 shows the total ion chromatogram (TIC) for the studied VOC.

### 3.2. Volatile Blood and Room Air C-+Onstituents

As mentioned before, a total number of 38 VOC compounds were detected and identified in the measured blood and air inside the carpentry shop, which came from the solvent of the paint and from the traffic nearby. All of the measured concentrations of each compound for each person were measured, and the average of these measurements is summarized in Table 2. The most common compounds identified in the blood were hydrocarbons and ketones [38,39]. The observed concentrations ranged from 3 ng/L for trichloroethene to 91 ng L^−1^ for toluene and 270 ng L^−1^ for 2,4-diisocyanate, which came together with toluene and xylene from the paint solvents that have been used for the wood in the carpentry shop and from the paints on the walls. More than half of all quantified species (80%) exhibited mean concentration values below 50 ng L^−1^, which is the allowed value for most of the VOCs.

Despite the fact that the VOC blood levels reported in this study are from peripheral venous blood, comparing these values to room air concentrations can provide useful information on the origin (endogenous/exogenous) of specific volatile species. A high incidence in blood and low levels in room air, for example, may indicate that blood is the predominant source of an analyte in a breath. Exogenous pollutants, on the other hand, have high levels in the air but low levels in the blood.

Several substances were reported to be exclusively or in higher concentrations in the blood of smokers. However, because only seven smokers worked in the carpentry sector and were chosen for this study, the taxonomy of these species was validated using qualitative data from previous studies [40].

However, the high blood detection rate of these species indicates that at least some of their blood abundance is derived from foreign sources and that the similarity between breath and room air levels is most likely owing to an equilibration between blood and room environment.

A total of 24 hydrocarbons (HCs) were discovered in blood and/or breath samples, making this family the most prevalent chemical class in this investigation. This indicates that room air is a key source of these HCs in the blood. Several hydrocarbons were abundant in room air while being rather infrequent in blood. These hydrocarbons had larger quantities in room air than in the blood, and for some of the other ones, there was no statistically significant difference between room air and the blood. The low detection incidence of these HCs in the blood may be explained by their low blood solubility [41,42,43], meaning that the venous blood concentrations were most likely near the analytical limits of the used method. Low blood levels of several other HCs can likewise be explained by very low, close-to-LOD breath levels.

The residual HC detection rate was frequently less than 30%. A number of aromatic chemicals were measured in blood and breath samples. The most common chemicals were benzene, toluene, and styrene.

Benzene, toluene, and o-xylene were discovered to be smoking-related species, as well as paint solvents. For example, benzene levels in the blood and breath of smokers were 10 times higher than in nonsmokers (14–20 (18) ng L^−1^ and 9–14 (11.2) ng L^−1^, respectively).

Toluene levels differed similarly between smokers and nonsmokers (78–87 (83) ng L^−1^ vs. 82–98 (85) nmol L^−1^ for blood and 110–120 (115) ng L^−1^ for air). One possible source of this volatility in human organisms could be the paint solvent and the decomposition of specific chemicals [43].

Two aldehydes (2-propenal and propanal) were measured in blood samples in this investigation, with incidence rates of around 50%.

2-propenal levels were found to be modest in the study, ranging from 32 to 43 (39) ng L^−1^.

This volatility was found in both the blood and room air, with only a minor variation between the two.

Propanal was found to have lower blood levels (63–69 (65) ng L^−1^) and also very low in room air, indicating that this species in the blood is exogenous. This investigation also assessed three esters. Methyl acetate was discovered in the blood and air samples and was shown to be lower in the blood than in room air, which is consistent with previous research [41].

All blood samples were tested for N-butyl acetate. This poor detection rate appears to be related to lower blood levels of this molecule when compared to methyl acetate. Acetonitrile had blood concentrations in the ng L^−1^ range. Although this chemical was found in nonsmokers’ blood and exhaled air, its levels in smokers were much greater (52–63 (57) ng L^−1^ in smokers’ blood and 41–49 (44) ng L^−1^ in nonsmokers’ blood and (127) ng L^−1^ in the air). These acetonitrile levels are consistent with those seen in other research [44,45,46] in the blood of smokers. Despite the fact that human blood concentration data are few and were frequently gathered for a specific set of people (e.g., smokers, mechanically ventilated patients), a reasonable agreement could be reached.

The majority of the values in our analysis were within the World Health Organization’s recommended range (WHO).

## 4. Conclusions

The current research aims to provide an idea of some VOCs compounds that employees or residential occupants can inhale due to the solvents used for carpentry paints in the blood and ambient air of healthy volunteers who work every day at a carpentry shop and live in nearby houses. Researchers have lately conducted studies to better understand the relationship between VOC exposure and health impacts. Techniques for properly and precisely measuring these chemicals in the blood have been developed using techniques that are becoming more widely used in the analytical community.

Gas chromatography with mass spectrometric detection in conjunction with pre-concentration processes (SPME) was employed to achieve this goal. In total, 38 VOC chemicals were discovered in the air and blood of 30 healthy adults. The amounts found in the blood varied from 3 ng L^−1^ to 270 ng L^−1^ and in the air from 9 ng L^−1^ to 290 ng L^−1^. The compounds detected belonged to various chemical classes, with hydrocarbons being the most plentiful. Ketones, heterocyclic compounds, and aromatic molecules were also detected in significant quantities.

Despite the fact that this study only included a limited number of smokers, it was discovered that smoking was associated with various blood and breath molecules.

Unsaturated hydrocarbons (1,3-butadiene, 1,3-pentadiene, 2-butene), furans (2,5-dimethylfuran), and acetonitrile fall within this category.

It should be noted that the proposed classification of measured species into systemic (blood-borne) and exogenous volatiles is purely speculative, as some species may have several sources. The term “blood-borne” does not always refer to a thing that is metabolic in nature. It also includes species associated with nutrition or drugs. Blood and blood-borne breath species with incidences greater than 80% are recommended human presence markers that will be confirmed in future field research. These approaches must be used with caution because contamination and analyte loss are still major concerns, but they are practical and have been used by numerous research groups. VOC levels in the body alter fast during exposure and after cessation of exposure. Most VOCs’ internal dose levels decline rapidly after exposure ends, with most having a half-life of a few hours; however, the actual decrease depends on the exposure scenario. Volatile organic compounds (VOCs) are a major public health concern throughout the developed world. Many critical aspects remain unanswered in determining exposure to these chemicals. Because VOCs are ubiquitous and very volatile, specialized procedures must be used in their analysis. The analytical methodology used to evaluate toxicants in biological materials must be adequately verified and meticulously carried out; inadequate quality assurance can result in inaccurate results that have a direct impact on treating exposed individuals. The pharmacokinetics of VOCs demonstrate that the majority of the internal dose of these compounds is rapidly cleared, while a portion is only slowly eliminated, and these chemicals may bioaccumulate. VOCs are found in the general population in the high parts-per-trillion range, but some people with much higher levels have apparently been exposed to VOC sources away from the workplace. Smoking is the most significant confounder to internal dose levels of VOCs and must be considered when evaluating suspected cases of exposure.

## 5. General Recommendations and a Warrens

As worldwide awareness of air pollution grows, so does the need for evidence-based recommendations for mitigation solutions.

While public policy plays an important role in lowering air pollution, personal choices can also help.

Limiting physical exertion outdoors on high air pollution days and near air pollution sources, reducing near-roadway exposure while commuting, using air quality alert systems to schedule activities, and using facemasks in prescribed circumstances are all supported by qualified evidence.

Avoiding cooking with solid fuels, ventilating and separating cooking spaces, and employing portable air cleaners equipped with high-efficiency particulate air filters are among more measures. We detail recommendations to help providers and public health officials advise patients and the public on personal-level strategies to reduce the risk posed by air pollution while also acknowledging the urgent need for well-designed prospective studies to better establish and validate interventions that benefit respiratory health in this context.

## Figures and Tables

**Figure 1 ijerph-20-05613-f001:**
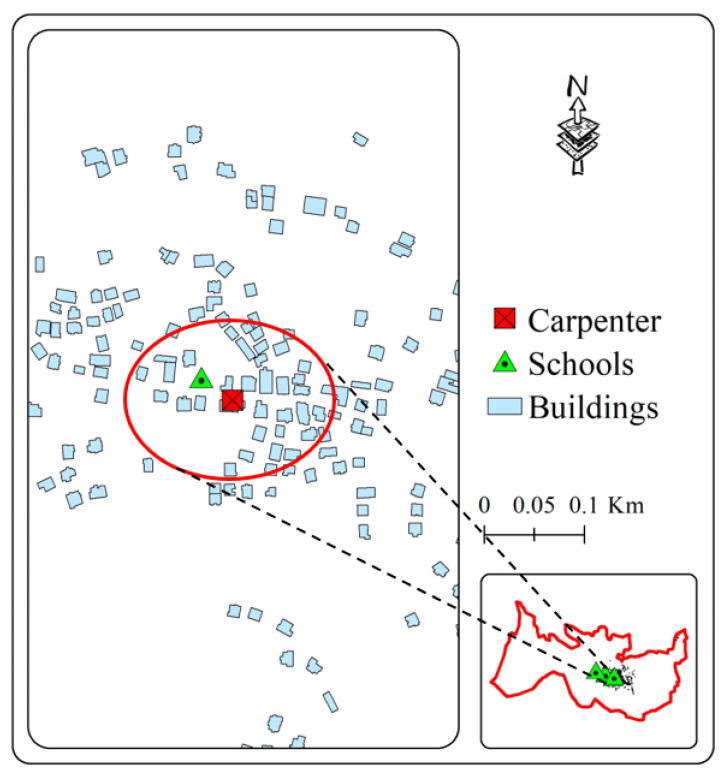
The study area with a focus on the selected school and carpenter neighborhood (the red zone).

**Figure 2 ijerph-20-05613-f002:**
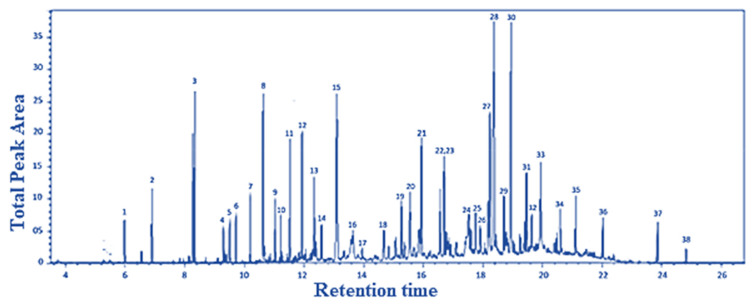
Total ion chromatogram (TIC) for the studied VOC detected in one of the studied samples showing all VOCs in Table 2 as a function of retention time.

**Table 1 ijerph-20-05613-t001:** List of volunteers who were chosen for the study.

Employee
1- XSQ-M-NS-26
2- LMQ-M-S-25
3- RMQ-M-S-32
4- AMM-M-S-37
5- LKM-M-S-49
6- JKM-M-NS-38
7- JMM-M-NS-53
8- SLM-M-S-51
9- KRM-M-NS-43
10- KLM-M-NS-40
Residence
11- SWQ-M-S-56
12- MSQ-M-S-28
13- KSQ-M-NS-17
14- SWJ-M-NS-62
15- RAQ-F-NS-60
16- SIQ-F-NS-50
17- RSQ-F-NS-28
18- ASQ-F-NS-25
19- ZMQ-M-NS-37
20- RMQ-F-NS-32
Students
21- LMQ-M-NS-12
22- JMM-M-NS-13
23- SKM-M-NS-15
24- RMM-M-NS-15
25- BRM-M-NS-14
26- BJQ-M-NS-12
27- LLQ-M-NS-15
28- LBQ-M-NS-15
29- QMM-M-NS-16
30- JMM-M-NS-15

XXX: name father family; M: Male; F: female; S: smoker; NS: non-smoker.

**Table 2 ijerph-20-05613-t002:** Analytical parameters for the determination of selected VOCS [1,2,3,4,5].

Retention Time	Analyte	Structure	Quant (m/z)	Average Value in Blood (ng/L) Mean ± SD	Average Value in Air (ng/L) Mean ± SD	MDL (ug/L) (GC)	Permissible Limit
5.98	acetonitrile	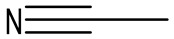	41	57 ± 0.105	127 ± 0.214	0.031	4.1 mg/day
6.95	ethyl acetate	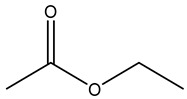	88.11	65 ± 0.113	87 ± 0.193	0.5	400 mg/L
8.24	1,1-Dichloroethene	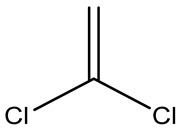	96	13.6 ± 0.081	25.6 ± 0.094	0.018	0.03 mg/L
9.13	2-propenal	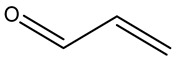	56	39 ± 0.092	74 ± 0.113	0.001	0.1 mg/L
9.26	Methylene chloride	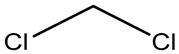	84	32.6 ± 0.088	38.5 ± 0.098	0.089	0–0.5 mg/kg body weight
9.54	Transe-1,2-Di Dichloroethene	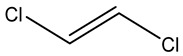	96	14.7 ± 0.076	32.4 ± −084	0.014	6.0 mL/kg body weight
10.28	propanal	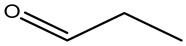	58	65 ± 0.121	83 ± 0.203	0.04 mg/L	0.104 mg/dL
10.75	Methyl tert-butyl ether	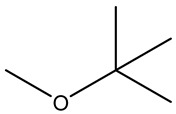	73	22 ± 0.072	51 ± 0.121	0.01	NA
11.41	Methyl acetate	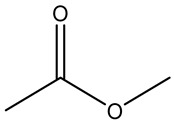	74	37 ± 0.082	61 ± 0.121	0.03	200 mg/L/10 h
11.75	cis-1,2-Dichloroethene	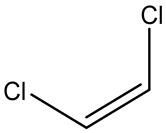	96	13.5 ± 0.056	17.2 ± 0.078	0.013	0.07 mg/L
11.90	chloroform	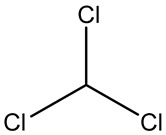	83	14.8 ± 0.065	18.2 ± 0.084	0.05	0.07 mg/L
12.65	1,2-Dichloroethane	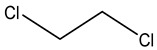	62	13.8 ± 0.072	16.5 ± 0.075	0.012	0.7 mg/m^3^
12.84	1,1,1-Trichloroethane	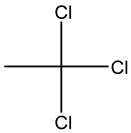	97	28 ± 0.092	28.5 ± 0.104	0.01	2000 μg/L
13.13	Carbon tetrachloride	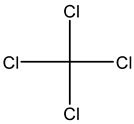	117	13 ± 0.084	22 ± 0.095	0.0 19	5 ppb
13.81	Benzene	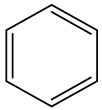	78	18 ± 0.093	28 ± 0.117	0.05	1 mg/L
13.96	Dibromomethane	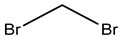	174	29 ± 0.084	47 ± 0.201	0.044	10 mg/L
14.68	1,2-Dichloropropane	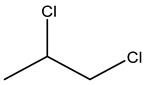	63	9.4 ± 0.103	14 ± 0.053	0.008	1.2 μg/m^3^
14.98	Trichloroethene	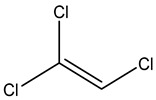	130	3 ± 0.004	7.5 ± 0.002	0.01	100 mg/L
15.72	Bromodichloromethane	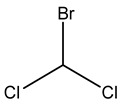	83	9 ± 0.009	4.7 ± 0.001	0.183	0.08 mg/L
15.91	Methyl proprionate	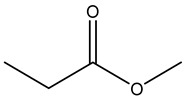	88	25 ± 0.08	81 ± 0.243	0.25 mg/L	200 mg/100 mL
16.79	2,5-Dimethylfuran	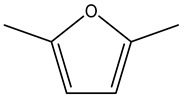	96	9 ± 0.009	4.8 ± 0.001	0.23	NA
16.82	1,1,2-Trichloroethane	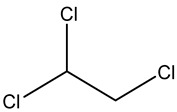	97	9.5 ± 0.008	12.8 ± 0.003	0.016	NA
17.14	n-butyl acetate	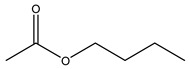	116.16	37 ± 0.072	42 ± 0.094	0.31	150 mg/L
17.66	Toluene	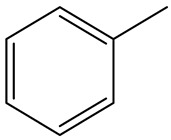	91	83 ± 0.321	115 ± 0.386	0.03	5 mg/L
17.86	Dibromochloromethane	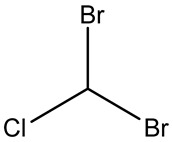	129	47.9 ± 0.075	148 ± 0.412	0.01	100 mg/L
18.22	Tetrachloroethene	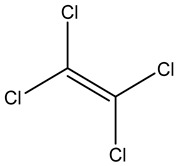	166	19 ± 0.045	26.4 ± 0.053	0.03	100 mg/L
18.31	Chlorobenzene	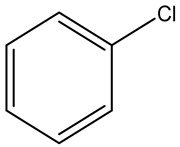	112	8.8 ± 0.008	15.2 ± 0.019	0.007	62 mg/L
18.68	Ethylbenzene	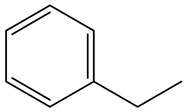	91	14 ± 0.104	16.7 ± 0.021	0.02	125 mg/L
18.83	m/p-Xylene	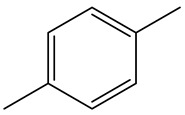	91	68 ± 0.328	123 ± 0.331	0.033	150 mg/L
19.43	Bromoform	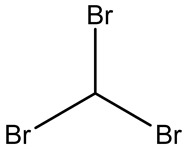	173	23 ± 0.129	38 ± 0.189	0.027	0.5 mg/L
19.62	Styrene	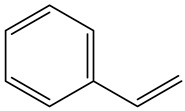	104	29 ± 0.183	118 ± 0.326	0.1	200 mg/L
19.74	1,1,2,2-Tetrachlroethane	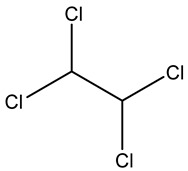	83	7 ± 0.005	26 ± 0.217	0.008	NA
20.63	o- Xylene	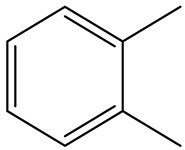	91	38 ± 0.213	53 ± 0.316	0.08	9 mg/L
21.27	1,3-Dichlorobenzene	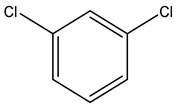	146	33 ± 0.141	48 ± 0.327	0.019	NA
22.01	1,2-Dichlorobenzene	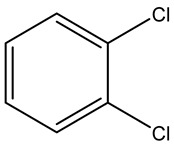	146	32 ± 0.170	48 ± 0.214	0.044	50 mg/l
22.85	1,4-Dichlorobenzene	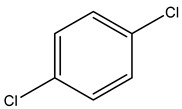	146	28 ± 0.140	42 ± −0.173	0.073	75 mg/L
23.15	toluene 2,4-diisocyanate	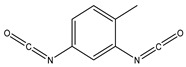	174.2	270 ± 2.73	291.5 ± 2.94	0.02	0.005 mg/L
24.62	Hexachloroethane	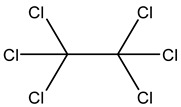	201	6.5 ± 0.004	13.4 ± 0.007	0.001	1 mg/L

Limits of detection (LODs) were calculated using extracted ion chromatograms and the standard deviation of 10 consecutive blank signals.

## Data Availability

The data used to support the findings of this study are available from the corresponding author.

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
