# Peer review of "Method Development for Detecting Low Level Volatile Organic Compounds (VOCs) among Workers and Residents from a Carpentry Work Shop in a Palestinian Village"

_ijerph, 2023, doi:10.3390/ijerph20095613_

Round 1

Author Response

Reviewer 1

Dear Reviewer

First of all let me thank you a lot for your comments. The paper looks different completely from the one I submitted after your comments. It looks to me we changed everything in the entire paper. I really appreciate your time and efforts.

International Journal of Environmental Research and Public Health: Monitoring and risk assessment of volatile organic compounds (VOCs) among workers and residence from a carpentry workshop in a Palestinian village.

Brief Summary: The authors present a study investigating the health risk of volatile organic compound (VOC) exposure on workers, residents, and students either working, living, or studying in/nearby of a carpentry shop in a Palestinian village. Personal exposure devices like portable passive monitors and air-collected samples were used to measure the VOC exposure at the individual level in the three abovementioned occupational groups along with an analytical method that was used to measure the level of 38 VOCs in the blood samples of the volunteers. The detection limit of the analytic method ranged from 0.001 to 0.15 ng/L and toluene diisocyanate as well as butyl cyanate was especially quantified among the 38 VOC compounds.

General concept comments Article: The research measures the level of different VOCs in the blood of different groups of people and in ambient air near a carpentry shop in a Palestinian village. The sample size of the study is relatively smaller to statistically conclude a significant association between VOC exposure and health risk. Furthermore, no risk assessment and related statistical analysis are done to assess the human health impacts of VOC exposure.

Review: Although the research is important in this field, the paper is poorly written, difficult to follow in some sections, and not well-organized. The overall delivery of the aim, scope, and findings of the study is moderately off and there are major grammatical issues throughout the paper. Authors are strongly suggested to take help from an expert or use web-based tools to correct all the grammatical errors. Specific concerns and questions needed to be clarified to improve the overall quality and readability of the paper. Pointwise comments are provided below:

Specific comments

Title

  • Pp. 1 – The word “residence” should be replaced with “residents”.
  • Done

Abstract

  • Pp. 1 – A brief summary of the key findings indicating the levels of detected VOCs should be included in the abstract.
  • Done

Introduction

  • Pp. 1 (lines 35-36) – The global and regional VOC sources and emissions in the indoor and outdoor environment and their impacts on the air quality should be discussed with appropriate references.
  • New paragraph with references has been added.

  • • 1 (lines 37-42) – The mechanism of how VOC emission affects human health should be elaborated to establish the necessity of the current study along with the quantitative findings from the previous studies.
  • New paragraph was added with reference.
  • Pp. 1 (lines 44-45) – The “The rate of VOC..” line is either written incorrectly or misrepresented.
  • It was removed since we added previous paragraphs describing the emission of VOCs.
  • Pp. 2 (line 73) – Authors should not address readers directly. A passive voice is suggested. done
  • Pp. 2 (line 79) – Authors should include how the limitations of the previous studies contributed to the already existing findings and how the current study is addressing and reducing those limitations as well as improving the outcomes.

The most difficult aspect of assessing VOCs is collecting samples and preparing samples, particularly VOC in blood.Gas chromatography Massspec is being used for analysis. However, GC-MS has two fundamental limitations: first, only a limited range of volatile, thermally stable chemicals may be analyzed, and second,  mass spectra frequently lack or have weak molecular ions.

A photoionization detector, or PID, is the most frequent equipment used by professionals to measure VOCs in a property. These instruments are typically portable and estimate the overall amount of VOCs in the air. Although laboratory analysis is the most convincing method of measuring VOCs, it is not always the best option due to the cost and turnaround time. Several labs can assess a sample for a panel of hundreds of VOCs, which can be quite useful when the irritant is unknown or there are numerous target chemicals. Certain VOCs, such as formaldehyde, require specialized laboratory testing and are not detectable by panel analyses. Typically, samples are collected on sorbent tubes, passive badges, or in evacuated canisters. The air quality professional will determine the optimal sampling method and laboratory analysis based on the specific situation. GC-IMS detected a wide range of volatile chemicals, primarily oxygen-containing VOCs such as alcohols, fatty acids, aldehydes, and ketones with low odor threshold values. The fingerprint plot of IMS spectra revealed variations in the composition of VOCs, indicating changes in wastewater quality during the treatment process. The GC-IMS approach may provide an efficient profiling method for changes in inlet water and treatment process performance at WWTPs.

That is what we meant by past studies' limitations contributing to existing conclusions and how the new study is different and unique.  

Materials and Experimental

  • Pp. 3 (line 97) – Figure 1 is very poor in resolution and not in the publication standard. It must be replaced with a higher-resolution map that clearly shows only the study area including the carpenter’s workshop, residential buildings, the school building, and other geographical elements with proper legends. Done
  • Pp. 12 (line 71) – The typo in the temperature unit should be corrected. Done
  • Pp. 12 (line 83) – The total population of the study area along with important socio-demographic information should be included with the study sample to show the socio-economic status and the percentage of the total population the study sample represents.
  • I added a paragraph to sec 2.1
  • Pp. 12 (line 83) – Authors must disclose that the individuals are exclusive in each sample group i.e., the same person was not considered under two sample groups. Yes. Each catogery is completely different from the other and has been used once.
  • Pp. 12 (line 84) – The number of employees who smoke indicated in this line does not match the number indicated in Table 2. Yes. It was corrected to 5. Thanks
  • Pp. 12 – The methodological steps for quantitative analysis of human health risk assessment (delineated by EPA) of different VOC exposure associated with certain diseases like COPD, asthma, cardiovascular disease, etc. is missing here and must be included if authors decide to include keywords such as “risk assessment” in the article title.
  • We will not include risk assessment in our title of the study. The title has been changed.
  • Pp. 13 (line 108) – Table 2 shows no such information. Yes. The sentence was re written again.
  • Thanks

Results and Discussion

  • Pp. 15 (line 184 to line 199) – The majority of this portion (excluding the quantitative findings of the current study) should be in the “Materials and Experimental” section.
  • The section has been moved to materials and experiment section. Thanks
  • Pp. 15 (line 199) – The descriptive significance of Figure 2 is missing here. Done
  • Pp. 16 (lines 208-209) – The purpose of the reference inclusion on this line is not clear and should be revised with an explanation or removed. removed
  • Pp. 17 (line 270) – Authors should quantitively discuss the specific health risks of the participated individuals associated with the dominant VOC exposures that were detected in the current study in detail.

As you suggested we added a paragraph in the introduction talking about the risk of VOCs to human and in our study the majority of the values in our analysis were within the World Health Organization's recommended range (WHO).

  •  

Conclusion

  • Pp. 17 – Authors should include a detailed portion of the uncertainties, limitations, and future scope of the current study.
  • I added to the conclusion section a paragraph that might determine obsticles and difficulties to measurements of VOC s.

  • Pp. 17 – Authors are suggested to include tailored recommendations targeted for the study population as well as general recommendations for all readers on how to reduce indoor VOC exposure and associated health risk at the personal level.

I added new section called recommensations and a warness.

Reviewer 2 Report

1.       The title of the article does not match the content. The authors evaluated an analytical method where 38 VOCs in blood samples at parts per trillion levels (ng/L) were detected using SPME-GC-MS. A pilot study was conducted  at a carpentry workshop in Palestine where they found common VOCs were toluene diisocyanate and butyl cyanate. The method development part was the main focus of the study and that was not reflected in the title.

2.       The abstract is incomplete. There is no information on results and discussions and conclusions on study findings. Please rewrite the abstract last part.

3.       The authors cited many articles where researchers investigated VOC levels in blood samples (on p. 2 of the manuscript; references 12 – 17). If these are the cases then what’s new in their approach? What’s the weaknesses and knowledge gaps in previous studies and how this new research study is addressing those? These are unclear in the Introduction part. Is the lower limit of detections being the main point? I suggest the authors rewrite the last part of the Introduction.

4.       Since the pilot evaluation of the VOC analytical method was tested in a rural carpentry shop, I suggest authors adding a separate paragraph on the potential VOC sources and related health outcomes related to carpentry with adequate references. There are many other VOC sources in a rural area and how those were considered in the study is unclear too.

5.       The Methods section needs to be more detailed. The standard curve preparations and analytical methods are standard procedures and do not need to be detailed. Some

6.       Table 1 data should be presented in the results section since method development is the main focus.

7.       Section, 2.7: How the ethical approval for recruiting human subjects were handled? Please provide this information at the beginning of this section.

8.       Lines 215-217: The contrasting observations of VOCs present at high levels in blood and low levels in room air are not discussed well. Did anyone in previous studies report similar discrepancies. Please explain.

9.       Line 273: Correct the typo.

10.   Conclusions: The last paragraph should be a part of the Discussion section. You may consider a separate paragraph on study limitations.

Author Response

Reviewer 2

Comments and Suggestions for Authors

Dear Reviewer

I would like to thank you very much for your time and efforts. The paper looks much better after I added your comments and made all of your suggestions.

With big respect

  1. The title of the article does not match the content. The authors evaluated an analytical method where 38 VOCs in blood samples at parts per trillion levels (ng/L) were detected using SPME-GC-MS. A pilot study was conducted  at a carpentry workshop in Palestine where they found common VOCs were toluene diisocyanate and butyl cyanate. The method development part was the main focus of the study and that was not reflected in the title.

Method development for detecting low level of volatile organic compounds (VOCs) among workers and residence from a carpentry work shop in a Palestinian village.

  1. The abstract is incomplete. There is no information on results and discussions and conclusions on study findings. Please rewrite the abstract last part.

I included three new paragraphs from results and conclusion in the abstract.

  1. The authors cited many articles where researchers investigated VOC levels in blood samples (on p. 2 of the manuscript; references 12 – 17). If these are the cases then what’s new in their approach? What’s the weaknesses and knowledge gaps in previous studies and how this new research study is addressing those? These are unclear in the Introduction part. Is the lower limit of detections being the main point? I suggest the authors rewrite the last part of the Introduction.

A paragraph has been added as you suggested. The main difference between previous studies is the technique and low detection limit. The pharmacokinetics of VOCs show that most of the internal dose of these compounds is quickly eliminated, but there is a fraction that is only slowly removed, and these compounds may bioaccumulate. VOCs are found in the general population at the high parts-per-trillion range, but some people with much higher levels have apparently been exposed to VOC sources away from the workplace. Smoking is the most significant confounder to internal dose levels of VOCs and

must be considered when evaluating suspected cases of exposure. Care must be exercised when using these methods, because contamination and loss of analyte are still significant concerns, but these methods are feasible and have been performed by various research groups. The levels of VOCs in the body change rapidly upon exposure and following cessation of exposure. Internal dose levels of most VOCs decrease rapidly after exposure ceases, with most having a half-life of a few hours, but the actual decrease depends on the exposure scenario.

In HS-SPME, VOCs are adsorbed directly from the sample onto polymer-coated fused-silica fiber. The fiber is then removed and the VOCs are thermally desorbed and injected into a gas chromatograph.

For these reasons we developed our method and used the technique. Beside, this is the first study that has been done in Palestine.

  1. Since the pilot evaluation of the VOC analytical method was tested in a rural carpentry shop, I suggest authors adding a separate paragraph on the potential VOC sources and related health outcomes related to carpentry with adequate references. There are many other VOC sources in a rural area and how those were considered in the study is unclear too.

New paragraph has been added as you suggested. Thanks

Organic chemicals are widely used as ingredients in household products. Paints, varnishes and wax all contain organic solvents, as do many cleaning, disinfecting, cosmetic, degreasing and hobby products. Fuels are made up of organic chemicals. All of these products can release organic compounds while you are using them, and, to some degree, when they are stored.

Chloroform can cause persistent liver toxicity. Benzene, an important petrochemical product and paint solvent, can decrease the number of red blood cells, leading to anemia and DNA breakage.[X} Toluene can be metabolized to toxic compounds; 5% of its metabolites are oxidized to benzaldehyde and cresols {Y}. Most of the reactive products are detoxified by conjugation to glutathione but the remainder may severely damage cells {[z]. Ethylbenzene and xylenes, important petrochemicals, can irritate the nose and throat upon inhalation, producing symptoms of central nervous system depression and affecting the liver and kidneys [Z].

  1. The Methods section needs to be more detailed. The standard curve preparations and analytical methods are standard procedures and do not need to be detailed. Some

I agree with you but if you do not mind the other reviewer asked some comments about these and he asked to add calibration curves. Also, most of paper includes these sections to help other people when do these studies to follow them. Thanks a lot.

  1. Table 1 data should be presented in the results section since method development is the main focus.

Table. 1 has been removed to results. Thanks

  1. Section, 2.7: How the ethical approval for recruiting human subjects were handled? Please provide this information at the beginning of this section.

This study based on the approval of the families of the students in the elementary school. Also, special form was filled and approved by the Institutional medical board of the university.

A paragraph was added as you suggested in sec 2.7.

  1. Lines 215-217: The contrasting observations of VOCs present at high levels in blood and low levels in room air are not discussed well. Did anyone in previous studies report similar discrepancies. Please explain.

In that paragraph I just explained definition only and I did not mean our results were higher in blood than in air. If this happened it can be just for smokers and the reason id smokers collected different VOCs in their blood on daily basis.

Smoking is the most significant confounder to internal dose levels of VOCs and

must be considered when evaluating suspected cases of exposure. Environ Health Perspect 104(Suppl 5):871-877 (1996)

  1. Line 273: Correct the typo. done
  2. Conclusions: The last paragraph should be a part of the Discussion section. You may consider a separate paragraph on study limitations.

Last paragraph has been added to the discussion section. New paragraph has been added as you suggested:

Care must be exercised when using these methods, because contamination and loss of analyte are still significant concerns, but these methods are feasible and have been performed by various research groups. The levels of VOCs in the body change rapidly

upon exposure and following cessation of exposure. Internal dose levels of most VOCs decrease rapidly after exposure ceases, with most having a half-life of a few hours, but the actual decrease depends on the exposure scenario.

Round 2

Reviewer 1 Report

Thank you for responding point-wise and making efforts to improve the overall quality of the manuscript. Although you have revised the manuscript to a good extent, I still have some concerns that needed to be resolved before I can recommend the manuscript for publication, 

1. Pp 4 (line 151): Authors have mentioned one school and one carpenter shop that was considered for the study but shown 4 schools and 3 carpenter shops in Figure 1 and haven't indicated which school and carpenter shop was selected for the study which is misleading.  

2. Pp 4 (lines 164-165): Figure 1 of the study area still lacks proper mapping and geographic features. Authors are suggested to provide a clear perspective mapping of figure 1 (a satellite map with author-included information on schools and shops may suffice). The authors need to revise the caption of figure 1 as well. 

3. Pp 8 (line 295): Why Table 2 is placed ahead of Table 1? The numbering order should be revised. 

Author Response

Comments and Suggestions for Authors

Dear Reviewer

Again, many thanks for your comments. I really appreciate your efforts and time for your valuable comments.

Thank you for responding point-wise and making efforts to improve the overall quality of the manuscript. Although you have revised the manuscript to a good extent, I still have some concerns that needed to be resolved before I can recommend the manuscript for publication, 

  1. Pp 4 (line 151): Authors have mentioned one school and one carpenter shop that was considered for the study but shown 4 schools and 3 carpenter shops in Figure 1 and haven't indicated which school and carpenter shop was selected for the study which is misleading. Thanks. A circle was made around the study area.  
  2. Pp 4 (lines 164-165): Figure 1 of the study area still lacks proper mapping and geographic features. Authors are suggested to provide a clear perspective mapping of figure 1 (a satellite map with author-included information on schools and shops may suffice). The authors need to revise the caption of figure 1 as well. Many thanks. I made a mistake when I mentioned google earth. This map was designed from some figures for the village and modified by adding the lables. I wish I have google earth so I can show 3 D for the buildings but still we do not have funds to join google earth. I also changed the caption of the figure. I hope this map will be enough for the purpose of study.
  3. Pp 8 (line 295): Why Table 2 is placed ahead of Table 1? The numbering order should be revised. 

Done

Reviewer 2 Report

I have reviewed the responses from authors and the revised manuscript. The authors have adequately responded to my questions and concerns and significantly improved the manuscript. 

Author Response

Rev 2

Comments and Suggestions for Authors

Dear Reviewer

We really appreciate you for giving us the time and efforts for not only revision but also for improving the manuscript.

I have reviewed the responses from authors and the revised manuscript. The authors have adequately responded to my questions and concerns and significantly improved the manuscript. 

Submission Date

18 February 2023

Date of this review

26 Mar 2023 01:20:40
